# Climate-Smart Tillage Practices with Straw Return to Sustain Crop Productivity

Zhen Liu [1,2], Ning Wang [1], Jinling Lü [1], Lan Wang [1], Geng Li [1] and Tangyuan Ning [1,*]

1   State Key Laboratory of Crop Biology, Key Laboratory of Crop Water Physiology and Drought-Tolerance Germplasm Improvement, Ministry of Agriculture and Rural Affairs, College of Agronomy, Shandong Agricultural University, Tai'an 271018, China
2   Shandong Key Laboratory of Eco-Environmental Science for Yellow River Delta, Binzhou University, Binzhou 256603, China
*   Correspondence: ningty@163.com; Tel./Fax: +86-0538-8242653

**Abstract:** Climate change seriously threatens global crop production. However, there are few reports on field crop yield and yield components based on long-term different climate conditions. The objectives of this study were to identify and compare the differences in crop yield and yield components in long-term tillage and straw returning under different climate regions. Conventional tillage (CT) and rotary tillage (RT) in combination with no straw return and whole straw return (S) were conducted under a wheat (*Triticum aestivum*)–maize (*Zea mays*) cropping system in cool-wet and warm-dry regions from 2010 to 2019. We hypothesized that long-term suitable tillage under warm-dry or cool-wet regions can increase the yield and components of wheat and maize, and temperature and precipitation had significant effects on crop yield and yield components. Conventional tillage with straw return (CTS) in the warm-dry region and rotary tillage with straw return (RTS) in the cool-wet region can increase the yield and yield components of wheat and maize, respectively, compared with CT. The yield stability of wheat was higher than that of maize under the two climate conditions. Compared with tillage practices, the effects of experimental sites and straw return on crop yield and yield components were more remarkable. The combination of mean temperature, annual precipitation, and yield components explained 75% and 100% of the variance in the wheat yield and maize yield, respectively. The thousand-kernel weight was the key factor in regulating wheat yield, and kernel number was the key factor in regulating maize yield. In conclusion, the combination of rotary tillage in cool-wet regions or conventional tillage in warm-dry regions with straw return is a good technique for increasing crop security.

**Keywords:** climate trend; tillage; straw returning; crop sustainability



## 1. Introduction

In the past 100 years, the temperature of the Earth's surface has increased by 0.74 °C [1], and it will increase by 1.0–1.5 °C before 2100 [2]. Due to global warming, the number of regions with relatively warm and extremely dry weather may increase by the end of this century [3]. Climate change is very unstable and difficult to control by people through farmland management measures [4]. Climate change is altering water supplies worldwide and further threatening food productivity [5]. In the coming decades or even centuries, climate change will have a significant impact on the agroecological food system [6]. Due to climate change and global population growth, achieving food security on limited arable land is a major challenge of the 21st century [7]. Climatic vulnerabilities, especially heat waves, drought, or rainstorms occurring in key growth stages, such as anthesis and filling, will lead to serious yield loss [8]. For instance, relatively high temperatures during grain filling can decrease the grain filling rate [9], shorten the grain filling duration [10], and ultimately lead to a significant grain yield reduction. Globally, there is a negative correlation between climate warming and some crop productivity, such as wheat,

maize, and barley [11]. Future agriculture will be challenged by changes in the spatial and temporal distribution of precipitation [12]. Precipitation was identified as the most important determinant of many crop yields [13]. The interannual variation in wheat yield is very large due to the unstable seasonal climate pattern, especially the irregular precipitation distribution and high temperature during the grain filling period [14]. In the past 50 years, temperature and precipitation have increased in Northeast China, with reduced snow and increased spring drought [15]. Wheat production in northern Kazakhstan has decreased due to the combination of warming and drought since 1980 [16]. However, the results are often different or even opposite due to the differences in the duration of the experiment, soil type, management measures, and climatic conditions. As a result, the effectiveness of cropping systems is highly geographically specific, and the impact of yield constraints can vary greatly depending on environmental conditions and their interactions with management practices [17]. Crops need to adapt to changing climate conditions to guarantee food security. Understanding the link between climate change and crop productivity is therefore crucial to assessing the adaptability of agricultural systems to future climate changes [18].

Further improvement in crop yields is needed for future food security [19]. Winter wheat–summer maize rotation is the most typical cropping system on the North China Plain [20]. The main objective of the winter wheat–summer maize cropping system is to produce sufficient crop yields to sustain the population [21]. The precipitation in the wheat growing season accounts for 20–30% of the annual precipitation [22]. Low precipitation and high annual variation are the main reasons for the low and unstable wheat yield in this region [23]. Crop yield is determined by yield components, including ear density, kernel number, and thousand-kernel weight [24]. The genetic process of crop yield is mainly related to kernel number and kernel weight [25,26]. Yield and yield components are determined by genotype, cropping system, agronomic management, climatic conditions, and plant–plant interactions [27–29]. Therefore, reasonable climate-smart crop management is very important to sustain crop yield.

Conservation tillage, such as minimum or no-tillage combined with mulching and crop rotation, can reduce soil runoff, evaporation, erosion, and degradation [30]. A study found that minimum tillage combined with straw mulching is effective in reducing soil temperature and increasing soil moisture and crop yield [31]. Moreover, the combination of minimum or no-tillage and straw return can effectively promote soil macrocohesion, change the abuse of conventional tillage systems to consume soil nutrients, and increase soil fertility and crop yield [32]. Straw mulching works worldwide by absorbing the kinetic energy of raindrops to reduce soil compaction and erosion caused by precipitation [33]. Straw mulching and returning can conserve water by reducing surface loss and increasing infiltration, thereby improving the availability of soil water, thus increasing crop yield and water use efficiency [34,35]. Conservation tillage under well-drained soils can increase maize yield, while minimum tillage does not appear to significantly affect yield stability [13].

However, the effects of long-term tillage and straw return under different climate conditions on field crop yield and yield components are still unclear. The objectives of this study were: (1) to identify the differences in wheat and maize yields and yield components in long-term tillage and straw returning; (2) to compare the yield and yield component stability of wheat and maize under tillage and straw returning practices; (3) to clarify the correlation between crop yield, yield components, and mean temperature and precipitation. It was hypothesized that long-term suitable tillage under warm-dry or cool-wet regions can increase the yield and components of wheat and maize, and that temperature and precipitation have a significant effect on crop yield and yield components.

## 2. Materials and Methods

### 2.1. Experimental Site

One warm-dry region (Tai'an) and one cool-wet (Longkou) region of Shandong Province in northern China were selected in this study (Figure 1). This former and the latter have a temperate continental monsoon climate, with mean temperatures of 14.4 °C and

13 °C and mean annual precipitation values of 580.6 mm and 650 mm, respectively, during 2010–2019 (Figure 2). The experiment was first established in Tai'an (36°09′ N, 117°09′ E) in 2002. Considering the climatic factors, another experiment with the same treatments was conducted in Longkou (37° 38′ 30″ N, 120° 24′ 9″ E) in 2006. The soils in the two regions were the same and were classified as Cambisols (Food and Agriculture Organization). Only the temperature and precipitation were different. The major initial (2002) properties in the 0–20 cm soil layer in Tai'an were: 40% sand, 44% silt, and 16% clay; the pH was 7.09, and the soil contained 7.2 g kg$^{-1}$ soil organic carbon, 1.3 g kg$^{-1}$ total nitrogen, 0.8 mg kg$^{-1}$ available phosphorus, and 41.3 mg kg$^{-1}$ exchangeable potassium. The major initial (2006) properties within the 0–20 cm soil depth in Longkou were: 47.0% sand, 37.0% silt, and 15.0% clay; the pH was 6.4, and the soil contained 7.6 g kg$^{-1}$ soil organic carbon, 115.0 mg kg$^{-1}$ total nitrogen, 36.7 mg kg$^{-1}$ available phosphorus, and 91.2 mg kg$^{-1}$ exchangeable potassium.

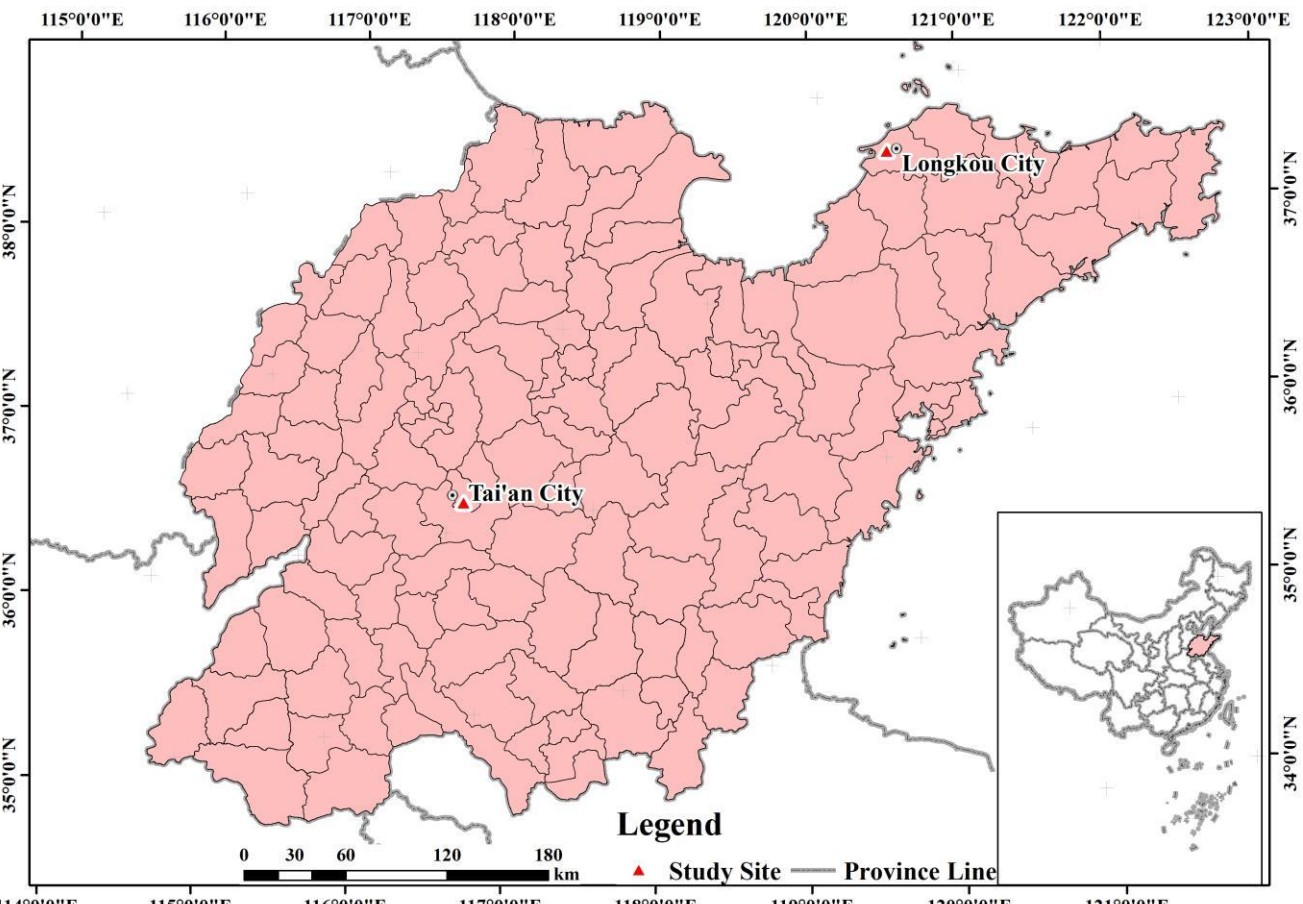

**Figure 1.** The locations of the Tai'an and Longkou experimental sites.

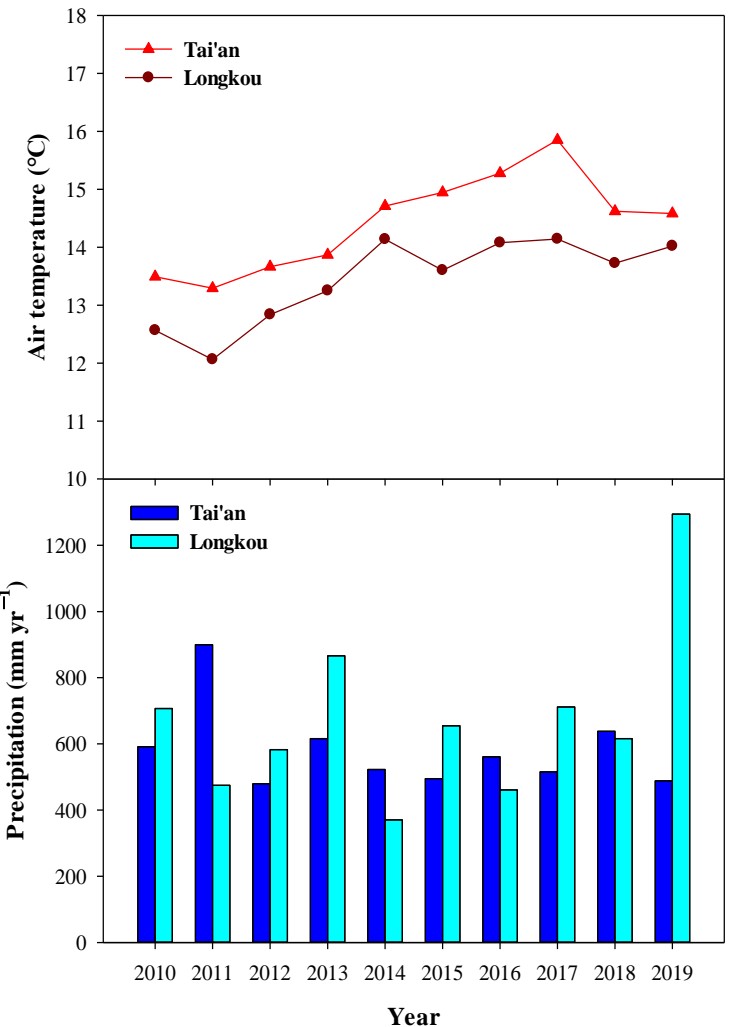

**Figure 2.** Temperature and precipitation of the Tai'an and Longkou experimental sites.

## 2.2. Experimental Design

A typical cropping system of winter wheat–summer maize was chosen at the Tai'an and Longkou experimental sites. Winter wheat was generally sown in mid-October and harvested in early June of the next year. Summer maize was sown in mid-June and harvested in early October. Fertilization application was carried out according to the local high-yield field. A 10-year (2010–2019) long-term experiment was conducted in both regions. Two tillage practices included conventional tillage (CT, 30 cm depth) and rotary tillage (RT, 10 cm depth). Four treatments were designed by two maize and wheat straw return methods, with no straw return or whole maize and wheat straw return (S). Three replicates were set for each treatment, and the size of each plot was $15 \times 4$ m$^2$ at Tai'an and $15 \times 15$ m$^2$ at Longkou. The operational procedures for conventional tillage were: maize was harvested mechanically, and straw was crushed (3–5 cm long) and returned to the field at the same time for straw return treatment (or after harvesting the maize manually, cut the straw against the ground and move it out of the farmland for no straw return treatment)–fertilizer application–stubble plowing with notched harrow–plowing with moldboard plow (25 cm)–rotary tillage with rotavator (10 cm)–wheat sowing–irrigation–herbicide application. The operational procedures for rotary tillage were: maize was harvested mechanically, and straw was crushed (3–5 cm long) and returned to the field at the same time for straw return treatment (or after harvesting the maize manually, cut the straw against the ground and move it out of the farmland for no straw return treatment)–fertilizer application–stubble plowing with notched harrow–rotary tillage with rotavator (10 cm)–

wheat sowing–irrigation–herbicide application. Rotary cultivator is an agricultural machine that breaks the soil by rotating the blade teeth. Tractors are often used as power sources for rotary cultivators to provide power for driving and operation.

### 2.3. Determination of Crop Yield and Components

The wheat grains within a 1 m$^2$ area were collected in each treatment with triplicates at maturity. The maize cobs of 5 m double rows were harvested from the center of each treatment with three replicates. The samples were air dried and threshed, and then ear density, kernel number, and thousand-kernel weight were measured to calculate the yield.

$$\text{Wheat grain yield (t ha}^{-1}) = \text{Ear density (ears ha}^{-1}) \times \text{kernel number per ear} \times \text{thousand-kernel weight (g 1000 kernels}^{-1})/10^9 \times (1 - \text{sample moisture content\%})/(1–13\%) \tag{1}$$

$$\text{Maize grain yield (t ha}^{-1}) = \text{Ear density (ears ha}^{-1}) \times \text{kernel number per ear} \times \text{thousand-kernel weight (g 1000 kernels}^{-1})/10^9 \times (1 - \text{sample moisture content\%})/(1–14\%) \tag{2}$$

### 2.4. Determination of Equivalent Ratios (Ers)

The relative values of the wheat and maize yield and components were standardized based on those in the CT, which was considered to be 1.

### 2.5. Determination of the Stability Index of Yield and Yield Components

The stability index (SI) is an important parameter for measuring the sustainability and stability of a system. The formula for calculating the SI is as follows [36]:

$$\text{SI} = (Y - \sigma)/Y_{max} \tag{3}$$

where Y represents the average yield or components per unit area during the experiment (t ha$^{-1}$), $\sigma$ is the standard deviation, and $Y_{max}$ is the largest crop yield or components among all experimental years (t ha$^{-1}$). The SI can be any value between 0 and 1; the greater the SI, the more stable the yield [37].

### 2.6. Statistical Analyses

Illustrations were prepared via Sigmaplot 12.5. The significance of differences between the treatments was analyzed using Duncan's test and independent-samples *t* tests at $p < 0.05$ and $p < 0.01$ via SPSS 18.0 Statistical Analysis System software [38]. Structural equation modeling (SEM) was used to evaluate the direct and indirect relationships between wheat and maize yield and mean temperature, annual precipitation, and yield components. SEM analysis was performed using Amos 21.0 (Amos Development Corporation, Chicago, IL, USA).

## 3. Results

### 3.1. Wheat Yield, Yield Components, and Ers Analysis

After the ten-year experiment, the mean ear density and kernel number ranged from 651.8 (RTS)–676.7 (10$^4$ ha$^{-1}$) (CTS) and 39.2 (RT)–41.4 (CT), respectively, at the warm-dry site (Figure 3A). The mean thousand-kernel weight and wheat yield of RTS increased by 10.2% and 6.8%, respectively; RT decreased the mean thousand-kernel weight and wheat yield by 1.3% and 8.5%, respectively, compared with CT (Table 1). The wheat yield and yield components, except for the thousand-kernel weight of rotary tillage, were lower than those of conventional tillage. Straw return increased the wheat yield and yield components compared with no straw application.

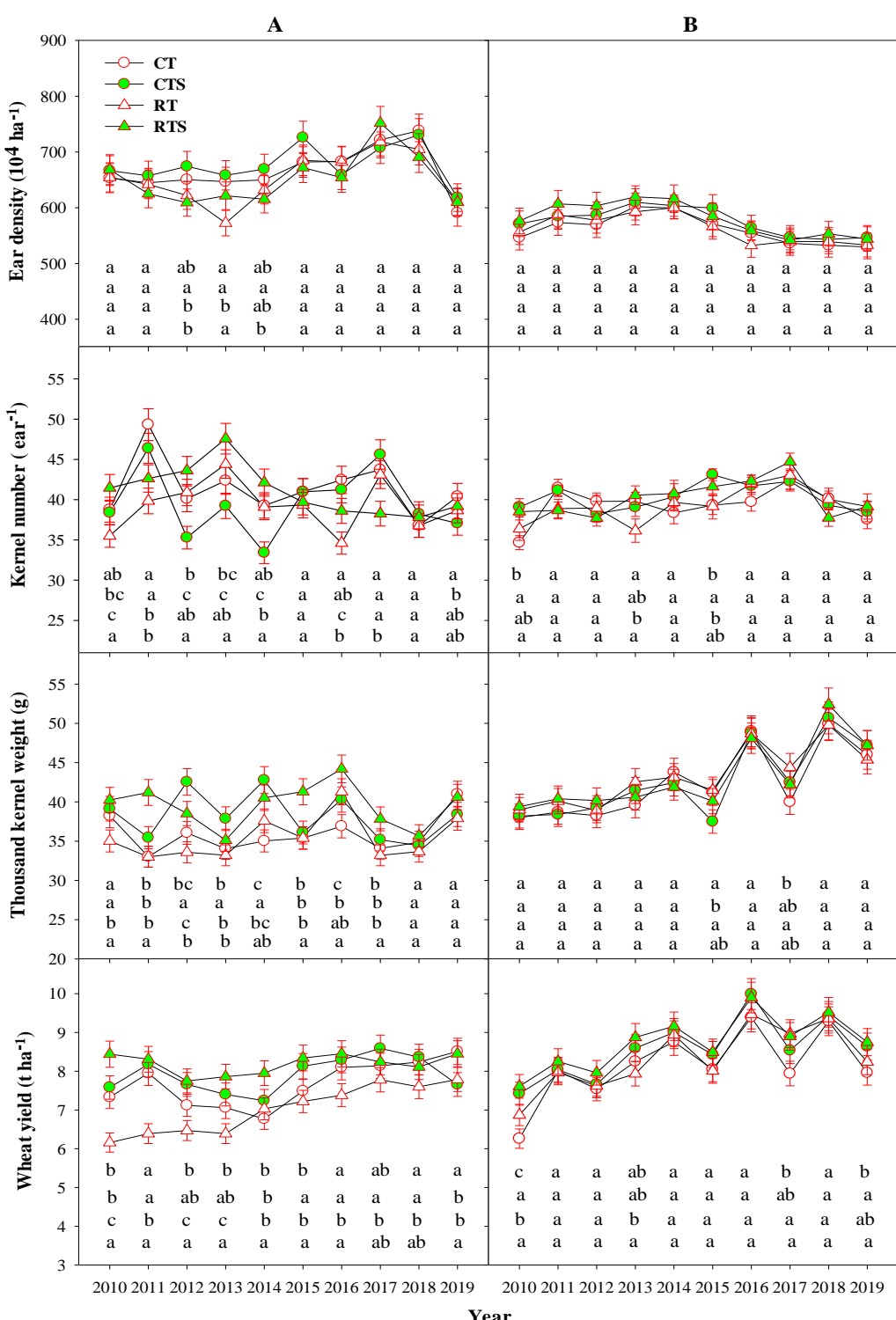

**Figure 3.** Wheat yield and yield components over time under tillage and straw returning practices at the Tai'an (**A**) and Longkou (**B**) experimental sites. CT: conventional tillage without straw returning; CTS: conventional tillage with straw returning; RT: rotary tillage without straw returning; RTS: rotary tillage with straw returning. T-like lines: standard deviation. Different letters in each column indicate significant differences between different treatments (*p* < 0.05; Duncan's test).

**Table 1.** Mean yield of wheat and maize under different tillage and straw returning from 2010 to 2019.

| | Warm-Dry Site | | Cool-Wet Site | |
|---|---|---|---|---|
| | Wheat Yield (t ha$^{-1}$) | Maize Yield (t ha$^{-1}$) | Wheat Yield (t ha$^{-1}$) | Maize Yield (t ha$^{-1}$) |
| CT | 7.7 a | 10.0 b | 8.1 a | 8.4 b |
| CTS | 7.9 a | 11.0 a | 8.6 a | 8.9 ab |
| RT | 7.0 b | 9.7 b | 8.3 a | 8.7 ab |
| RTS | 8.2 a | 10.3 ab | 8.7 a | 9.3 a |

CT: conventional tillage without straw returning; CTS: conventional tillage with straw returning; RT: rotary tillage without straw returning; RTS: rotary tillage with straw returning. Different letters in each column indicate significant differences between different treatments ($p < 0.05$; Duncan's test).

At the cool-wet site, compared with CT, the mean kernel number of CTS increased by 2.4%; the mean thousand-kernel weight of RT increased by 2.1%; and the ear density and wheat yield of RTS increased by 3.4% and 7.3%, respectively (Figure 3B; Table 1). The wheat yield and components except for kernel number under rotary tillage were higher than those under conventional tillage. Straw return increased the wheat yield and components compared with no straw application.

Box and whisker plots of the Ers of wheat yield and components were constructed for tillage and straw management at the warm-dry (Figure 4A) and cool-wet sites (Figure 4B). The median Ers of the ear density in the CTS was the largest at the warm-dry site (Figure 4A). The median Ers of the kernel number, thousand-kernel weight, and wheat yield in the RTS were the largest. The median Ers of the thousand-kernel weight and wheat yield in RT were the lowest. In addition to the Ers value of the kernel number in the CTS, the Ers value of the thousand-kernel weight in the RT, and the Ers value of the wheat yield in the RTS, the box plots for the other tillage and straw management practices were noticeably asymmetric. The comparative shortness of the box plots for the CTS and RT indicated that the Er values of the ear density and wheat yield were distributed similarly. The comparative tallness of the box plots for the RTS indicated that the Er values of the wheat yield and components showed wider distributions.

The median Ers of the ear density, kernel number, thousand-kernel weight, and wheat yield in the RTS were the largest at the cool-wet site (Figure 4B). The median Ers of the thousand-kernel weight and wheat yield in CT and RT were lower. There was no significant difference in these indexes among the four treatments. In addition to the Ers value of the wheat yield in the CTS, the box plots for the other tillage and straw management practices were noticeably asymmetric. The comparative shortness of the box plots for the CTS and RT indicated that the Er values of the ear density were distributed similarly. The comparative tallness of the box plots for the RTS indicated that the Er values of the kernel number showed wider distributions.

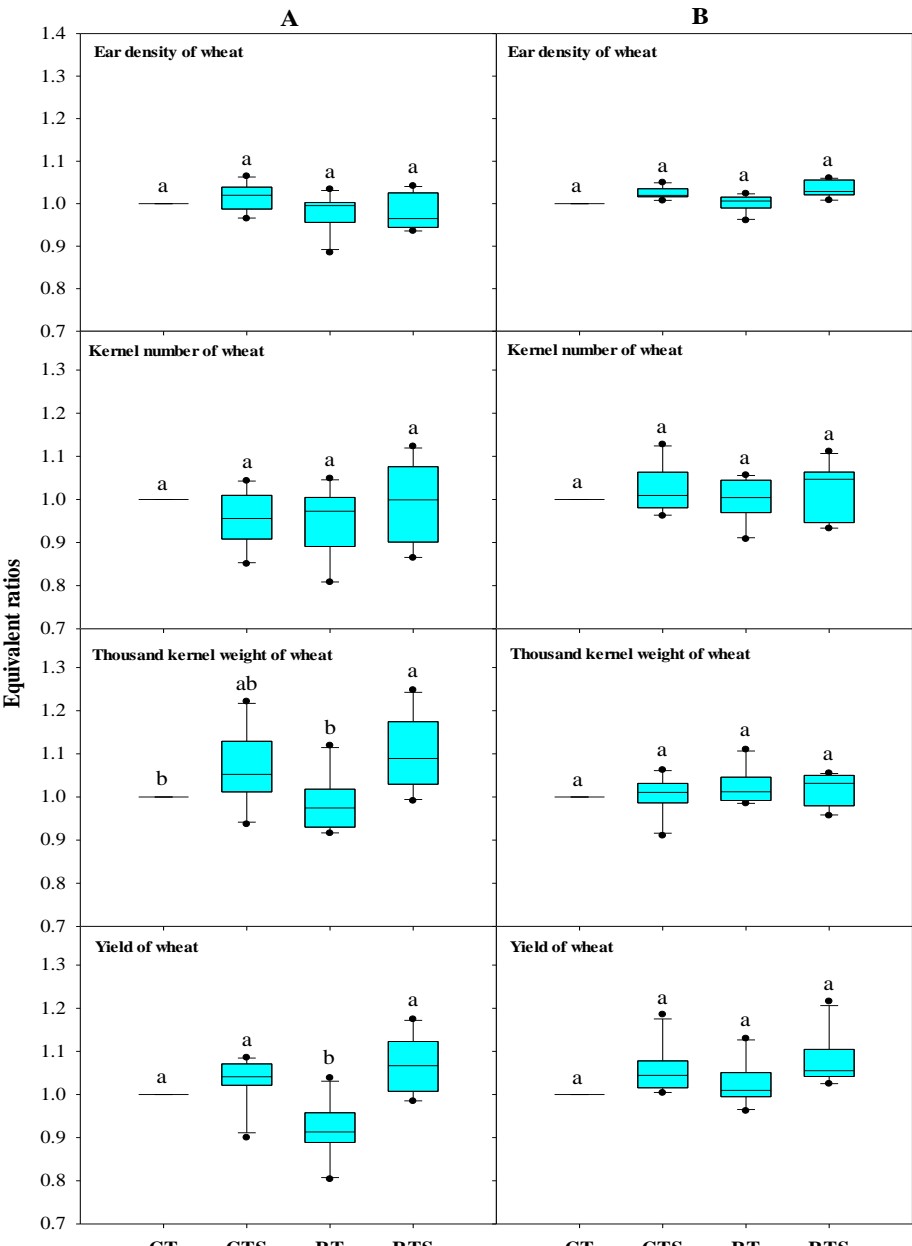

**Figure 4.** Box plots of equivalent ratios of the wheat yield and yield components under tillage and straw returning practices at the Tai'an (**A**) and Longkou (**B**) experimental sites. The horizontal line inside the boxes denotes the median value. The error bars show the upper/lower quartiles (1.5 interquartile distance). The line extending vertically from the box represents the variability beyond the range of the upper and lower quartiles, and the values represent the 10th and 90th percentiles, respectively; the points that lie outside of this range are shown as solid circles. CT: conventional tillage without straw returning; CTS: conventional tillage with straw returning; RT: rotary tillage without straw returning; RTS: rotary tillage with straw returning. Different letters in each group indicate significant differences between different treatments ($p < 0.05$; Duncan's test).

### 3.2. Maize Yield, Yield Components, and Ers Analysis

At the warm-dry site, compared with CT, the mean ear density, kernel number, thousand-kernel weight, and maize yield of CTS increased by 0.8%, 2.9%, 5.7%, and 10.2%, respectively; the mean ear density, kernel number, and maize yield of RT decreased by 4.3%, 0.6%, and 2.8%, respectively (Figure 5A; Table 1). The maize yield and components except for the thousand-kernel weight of rotary tillage were lower than those of conventional

tillage. Straw return increased the maize yield and components compared with no straw application. At the cool-wet site, compared with CT, the mean ear density, kernel number, thousand-kernel weight, and maize yield of RTS increased by 1.5%, 5.4%, 3.8%, and 10.9%, respectively; the mean ear density of RT decreased by 1.0% (Figure 5B; Table 1). The maize yield and components of rotary tillage were higher than those of conventional tillage. Straw return increased maize yield and components compared with no straw application.

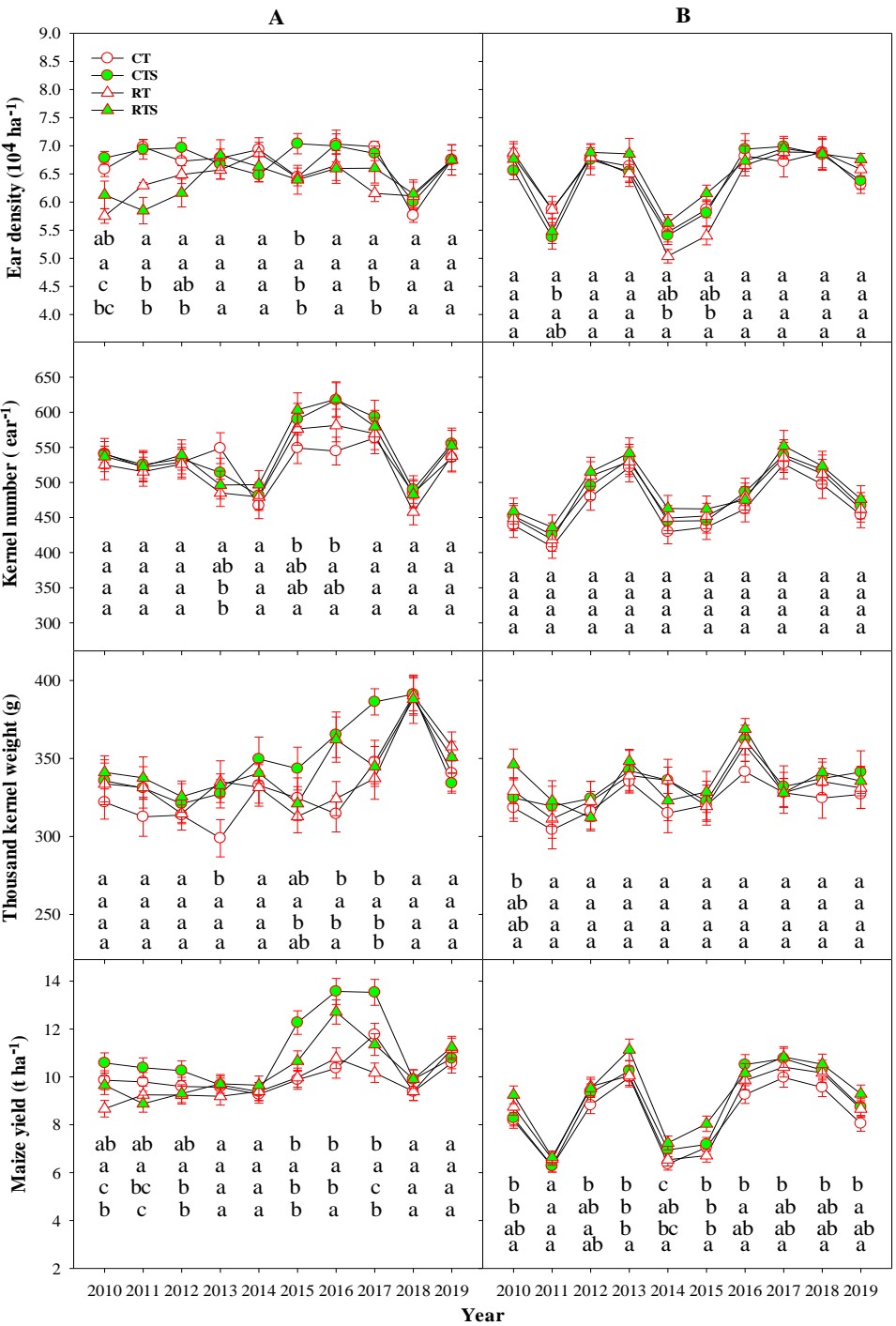

**Figure 5.** Maize yield and yield components over time under tillage and straw returning practices at the Tai'an (**A**) and Longkou (**B**) experimental sites. CT: conventional tillage without straw returning; CTS: conventional tillage with straw returning; RT: rotary tillage without straw returning; RTS: rotary tillage with straw returning. T-like lines: standard deviation. Different letters in each column indicate significant differences between different treatments (*p* < 0.05; Duncan's test).

Box and whisker plots of the Ers of the maize yield and components were constructed for tillage and straw management at the warm-dry (Figure 6A) and cool-wet (Figure 6B) sites. The median Ers of the ear density in the CT, kernel number in the RTS, thousand-kernel weight, and maize yield in the CTS were the largest at the warm-dry (Figure 6A) site. There were no significant differences in the median Ers of the ear density, kernel number, and thousand-kernel weight between these four treatments. The maize yield of CTS was significantly higher than that of CT and RT. The box plots for tillage and straw management were noticeably asymmetric. The comparative shortness of the box plots for the RT and RTS indicated that the Er values of the maize yield were distributed similarly. The comparative tallness of the box plots for the RT and CTS indicated that the Er values of the ear density and maize yield showed wider distributions, respectively.

The median Ers of the ear density, kernel number, thousand-kernel weight, and maize yield in the RTS were the largest at the cool-wet site (Figure 6B). There were no significant differences in the median Ers of the ear density, kernel number, and thousand-kernel weight between these four treatments. The maize yield of RTS was significantly higher than that of CT. In addition to the Ers value of the kernel number in the RT and thousand-kernel weight in the CTS, the box plots for the other tillage and straw management were noticeably asymmetric. The comparative shortness of the box plots for the CTS, RT, and RTS indicated that the Er values of the kernel number were distributed similarly. The comparative tallness of the box plots for the RT and CTS indicated that the Er values of the ear density and maize yield showed wider distributions, respectively.

From 2010 to 2015, there was a significant difference in wheat yield between straw return and no straw return, and there was no significant difference from 2016 to 2019, and there was a significant difference in maize yield only from 2015 to 2017 at the warm-dry site (Figure 7A). Wheat yield at the cool-wet site was significantly different only in 2010, and maize yield was significantly different only in 2014 and 2015 between straw return and no straw return (Figure 7B).

In addition, 2010 and 2014 were the eighth years of the warm-dry and cool-wet trials, respectively, and the comparison was conducted from 2010 to 2015 at the warm-dry site and from 2014 to 2019 at the cool-wet site. At the warm-dry site, in addition to RT, the wheat yield decreased at first (2010–2014), and then increased gradually, and the maize yield also showed the same trend under CT and CTS. At the cool-wet site, the wheat yield increased gradually, but it fluctuated greatly. The maize yield first increased (2014–2017), and then decreased (2018–2019). The changes in crop yield at the two sites were opposite.

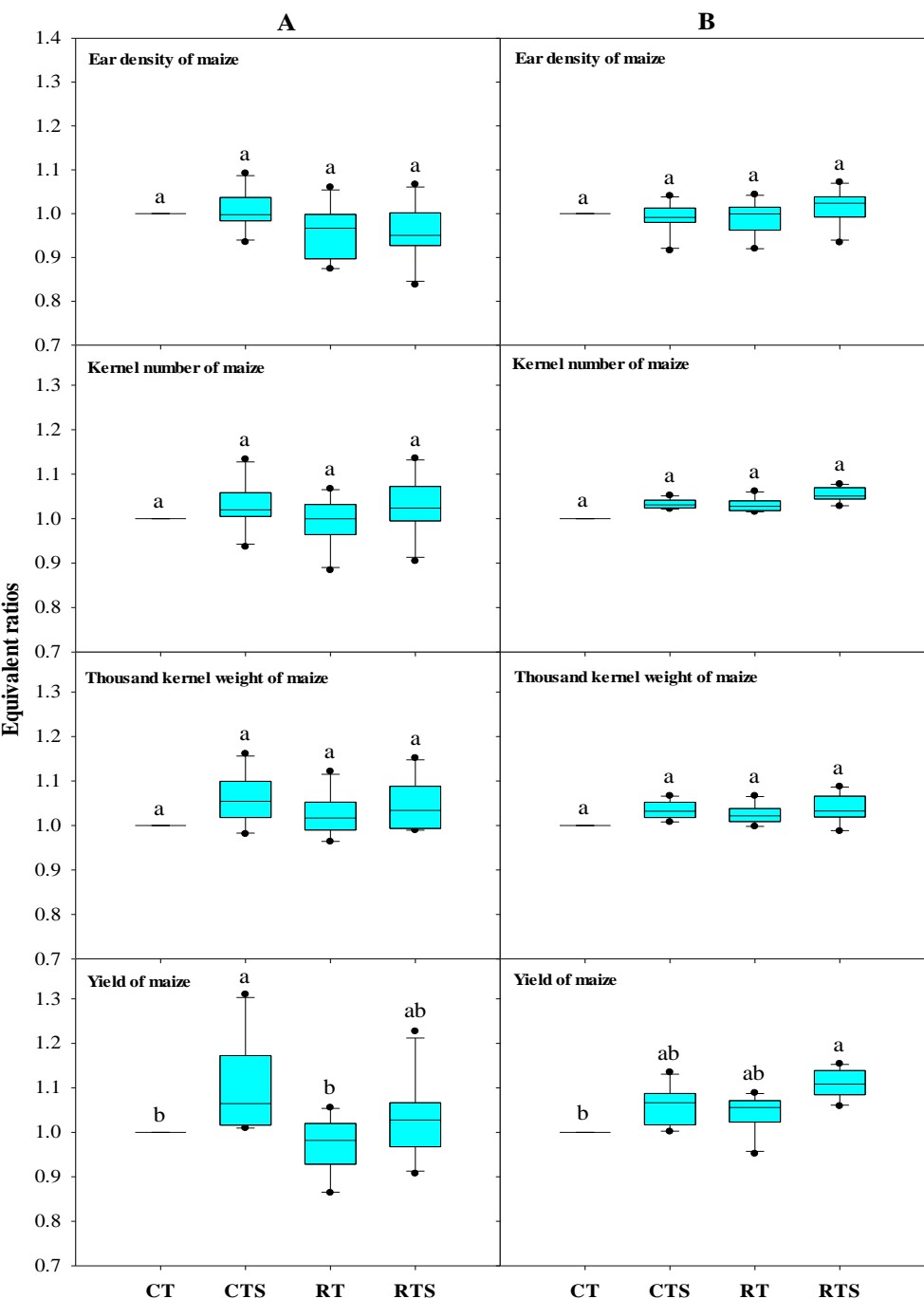

**Figure 6.** Box plots of equivalent ratios of the maize yield and yield components under tillage and straw returning practices at the Tai'an (**A**) and Longkou (**B**) experimental sites. The horizontal line inside the boxes denotes the median value. The error bars show the upper/lower quartiles (1.5 interquartile distance). The line extending vertically from the box represents the variability beyond the range of the upper and lower quartiles, and the values represent the 10th and 90th percentiles, respectively; the points that lie outside of this range are shown as solid circles. CT: conventional tillage without straw returning; CTS: conventional tillage with straw returning; RT: rotary tillage without straw returning; RTS: rotary tillage with straw returning. Different letters in each group indicate significant differences between different treatments (*p* < 0.05; Duncan's test).

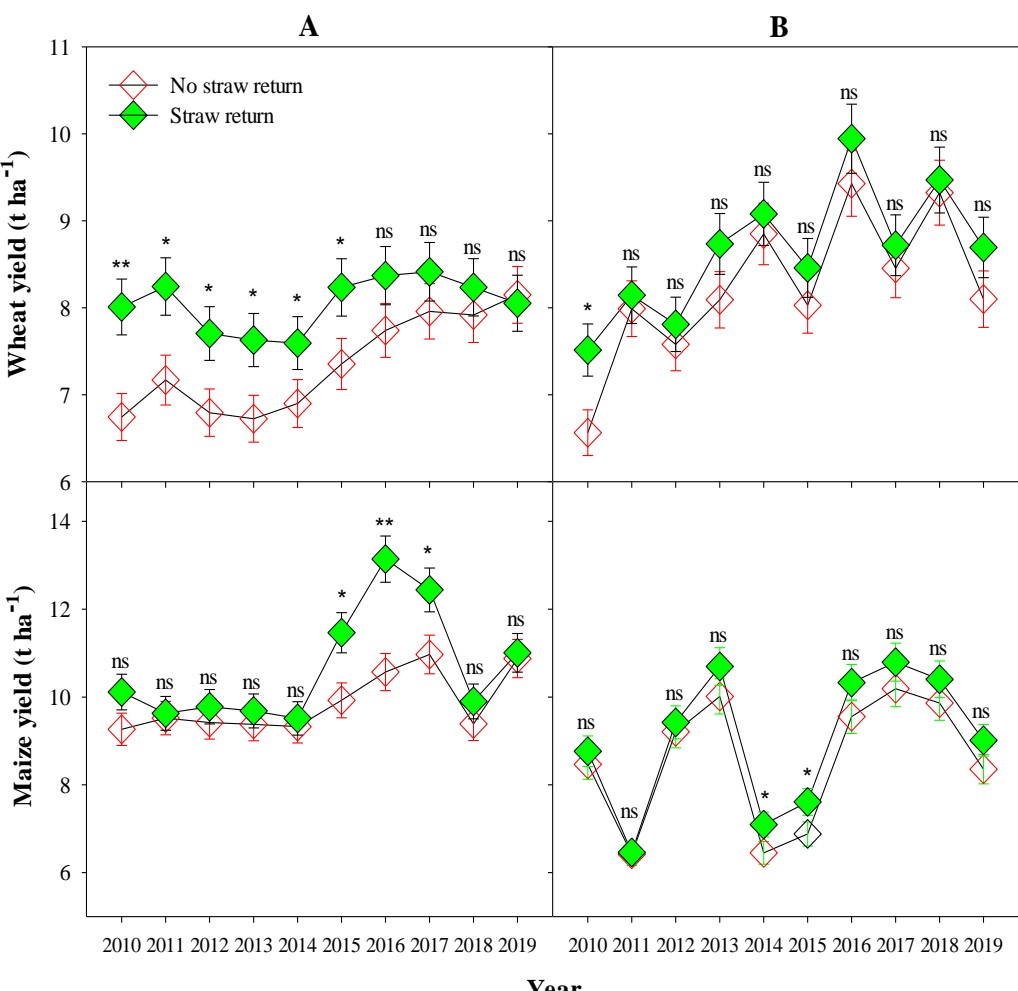

**Figure 7.** Wheat and maize yields over time under no straw return and straw return practices at the Tai'an (**A**) and Longkou (**B**) experimental sites. ** significant difference at 0.01 level; * significant difference at 0.05 level; ns: not significant difference.

### 3.3. Yield and Yield Component Stability of Wheat and Maize

The ear density and kernel number stability of maize were greater than those of wheat at the warm-dry site (Figure 8A). The thousand-kernel weight stability of wheat and maize was similar. The yield stability of wheat increased by 10.4% compared with that of maize. The ear density stability of wheat under CT was obviously higher than that under RTS, and the other yield components and yields of wheat and maize showed no obvious distinction under tillage and straw management. The ear density stability in wheat and maize and kernel number stability in maize under rotary tillage were lower than those under conventional tillage. The kernel number stability in wheat and yield stability in wheat and maize under rotary tillage were greater than those under conventional tillage.

The ear density and kernel number stability of wheat and maize were similar at the cool-wet site (Figure 8B). The kernel number and yield stability of wheat increased by 4.4% and 4.8%, respectively, compared with those of maize. The thousand-kernel weight stability of wheat was lower than that of maize. The yield components and yield of wheat and maize were not obviously different under tillage and straw management. The ear density and yield stability of maize under rotary tillage were greater than those under conventional tillage. The kernel number stability in wheat and thousand-kernel weight stability in maize under rotary tillage were lower than those under conventional tillage.

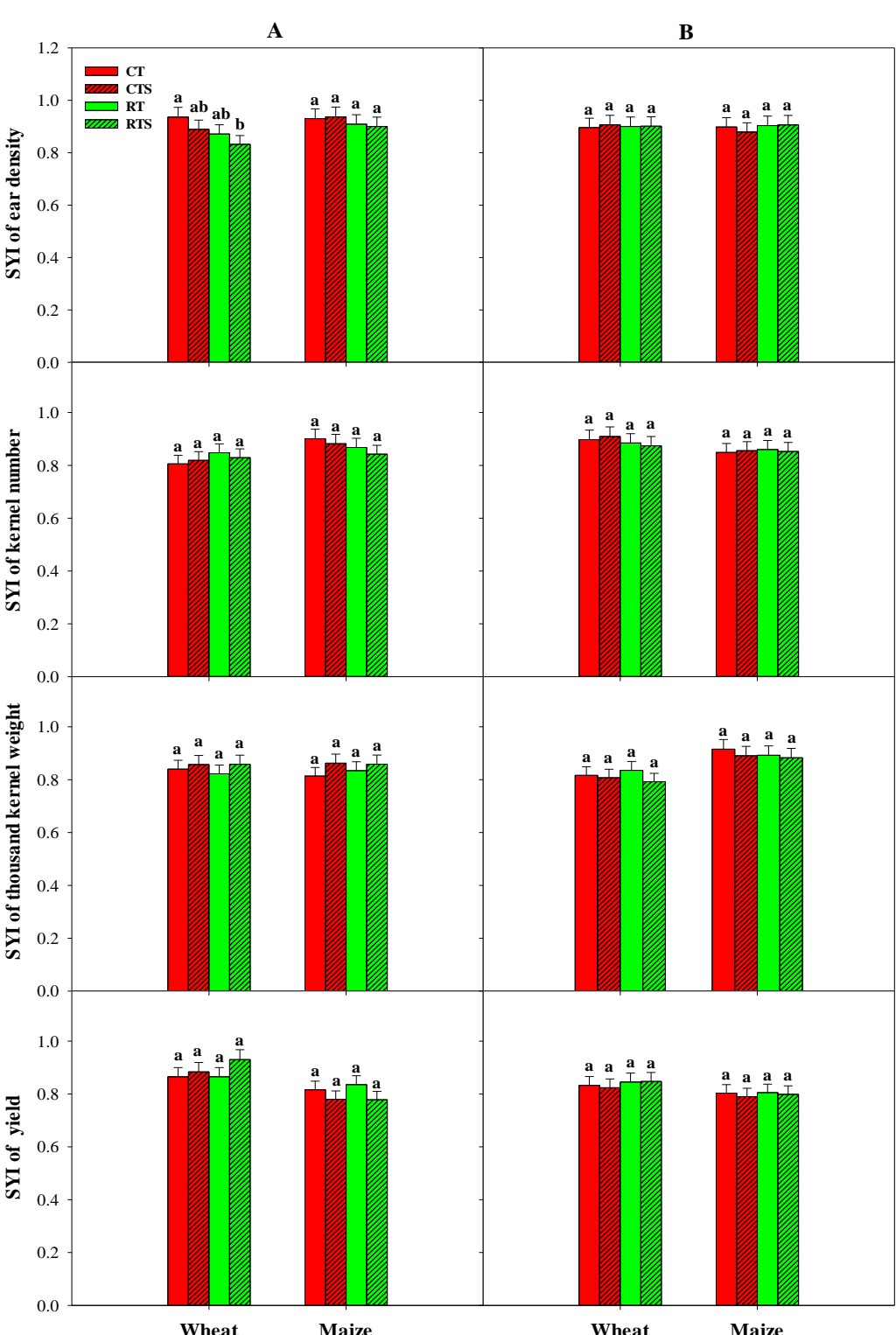

**Figure 8.** Yield and yield component stability of wheat and maize under tillage and straw returning practices at the Tai'an (**A**) and Longkou (**B**) experimental sites. CT: conventional tillage without straw returning; CTS: conventional tillage with straw returning; RT: rotary tillage without straw returning; RTS: rotary tillage with straw returning. T-like lines: standard deviation. Different letters in each group indicate significant differences between different treatments ($p < 0.05$; Duncan's test).

### 3.4. Correlation Analysis between Crop Yield and Components

There was an apparent negative correlation between ear density and wheat yield at the 0.01 level at the warm-dry and cool-wet sites (Table 2). There was a positive correlation between kernel number and wheat yield at the 0.05 level, thousand-kernel weight, and wheat yield at the 0.01 level. There were positive correlations between ear density, kernel number, thousand-kernel weight, and yield of maize at the 0.01 level.

**Table 2.** Correlation between crop yield and yield components.

| | Wheat Yield | Maize Yield |
|---|---|---|
| Ear density | −0.301 ** | 0.738 ** |
| Kernel number | 0.244 * | 0.867 ** |
| Thousand kernel weight | 0.724 ** | 0.460 ** |

*: Significant at $p < 0.05$; **: Significant at $p < 0.01$.

Under the conditions of long-term tillage practice, straw management, the experiment site, and their interaction had significant effects on the thousand-kernel weight of wheat (Table 3). Site and straw management had significant effects on wheat and maize yields. The site had significant effects on the ear density of wheat and the kernel number of maize. The interaction of site and tillage practice had a significant effect on maize yield. Tillage practice, the interaction of tillage practice and straw management, and the interaction of site, tillage practice, and straw management had no significant effects on crop yield or components.

**Table 3.** Three-way analysis of variance of the effects of long-term experiments on crop yield and yield components.

| Factor | Wheat | | | | | | | | Maize | | | | | | | |
|---|---|---|---|---|---|---|---|---|---|---|---|---|---|---|---|---|
| | Ear Density | | Kernel Number | | Thousand-Kernel Weight | | Yield | | Ear Density | | Kernel Number | | Thousand-Kernel Weight | | Yield | |
| | DF | *p* | DF | *p* | DF | *p* | DF | *p* | DF | *p* | DF | *p* | DF | *p* | DF | *p* |
| L | 1 | 0.000 | 1 | 0.417 | 1 | 0.000 | 1 | 0.000 | 1 | 0.153 | 1 | 0.000 | 1 | 0.121 | 1 | 0.000 |
| T | 1 | 0.431 | 1 | 0.773 | 1 | 0.385 | 1 | 0.989 | 1 | 0.215 | 1 | 0.555 | 1 | 0.580 | 1 | 0.727 |
| S | 1 | 0.311 | 1 | 0.500 | 1 | 0.023 | 1 | 0.001 | 1 | 0.716 | 1 | 0.095 | 1 | 0.074 | 1 | 0.001 |
| L × T | 1 | 0.285 | 1 | 0.828 | 1 | 0.800 | 1 | 0.186 | 1 | 0.111 | 1 | 0.404 | 1 | 0.779 | 1 | 0.011 |
| L × S | 1 | 0.589 | 1 | 0.517 | 1 | 0.027 | 1 | 0.292 | 1 | 0.908 | 1 | 0.835 | 1 | 0.630 | 1 | 0.376 |
| T × S | 1 | 0.858 | 1 | 0.187 | 1 | 0.580 | 1 | 0.106 | 1 | 0.707 | 1 | 0.937 | 1 | 0.427 | 1 | 0.516 |
| L × T × S | 1 | 0.711 | 1 | 0.163 | 1 | 0.440 | 1 | 0.089 | 1 | 0.543 | 1 | 0.844 | 1 | 0.842 | 1 | 0.470 |

L: experimental site; T: tillage practice; S: straw management; DF: degree of freedom; *p*: significance level.

### 3.5. The Correlation Analysis between Climate and Crop Yield

At the warm-dry site, there was a positive correlation between the ear density of wheat and the mean temperature in the wheat season at the 0.05 level (Figure 9). The difference between the wheat yield and mean temperature in the wheat season was nearly significant at the 0.05 level. The maize yield was positively correlated with the mean temperature in the maize season at the 0.01 level (Figure 10). At the cool-wet site, there was a positive correlation between the kernel number of wheat and the mean temperature in the wheat season at the 0.05 level (Figure 9). The difference between the wheat yield and mean temperature in the wheat season was nearly significant at the 0.05 level. The kernel number, thousand-kernel weight of maize, and maize yield were positively correlated with the mean temperature in the maize season at the 0.05 level (Figure 10). Other indices had no obvious divergence with mean temperature and precipitation.

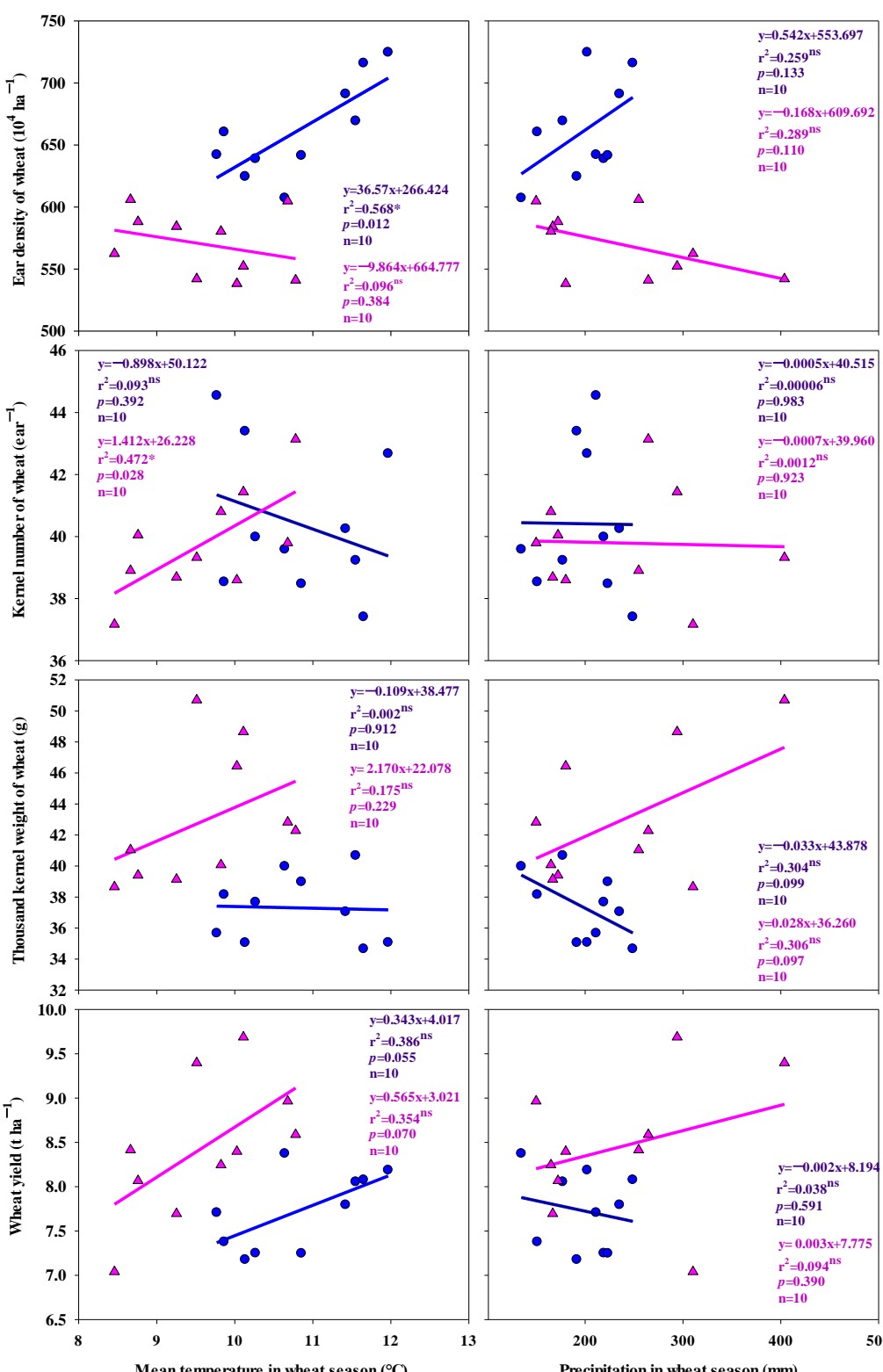

**Figure 9.** Correlation between wheat yield, yield components and mean temperature and precipitation in the wheat season. The blue circles, lines, and formulas represent the data of the Tai'an experimental site, and the pink circles and lines represent the data of the Longkou experimental site. r: correlation coefficient; *p*: significance level; n: sample size; *: Significant at *p* < 0.05. ns: not significant.

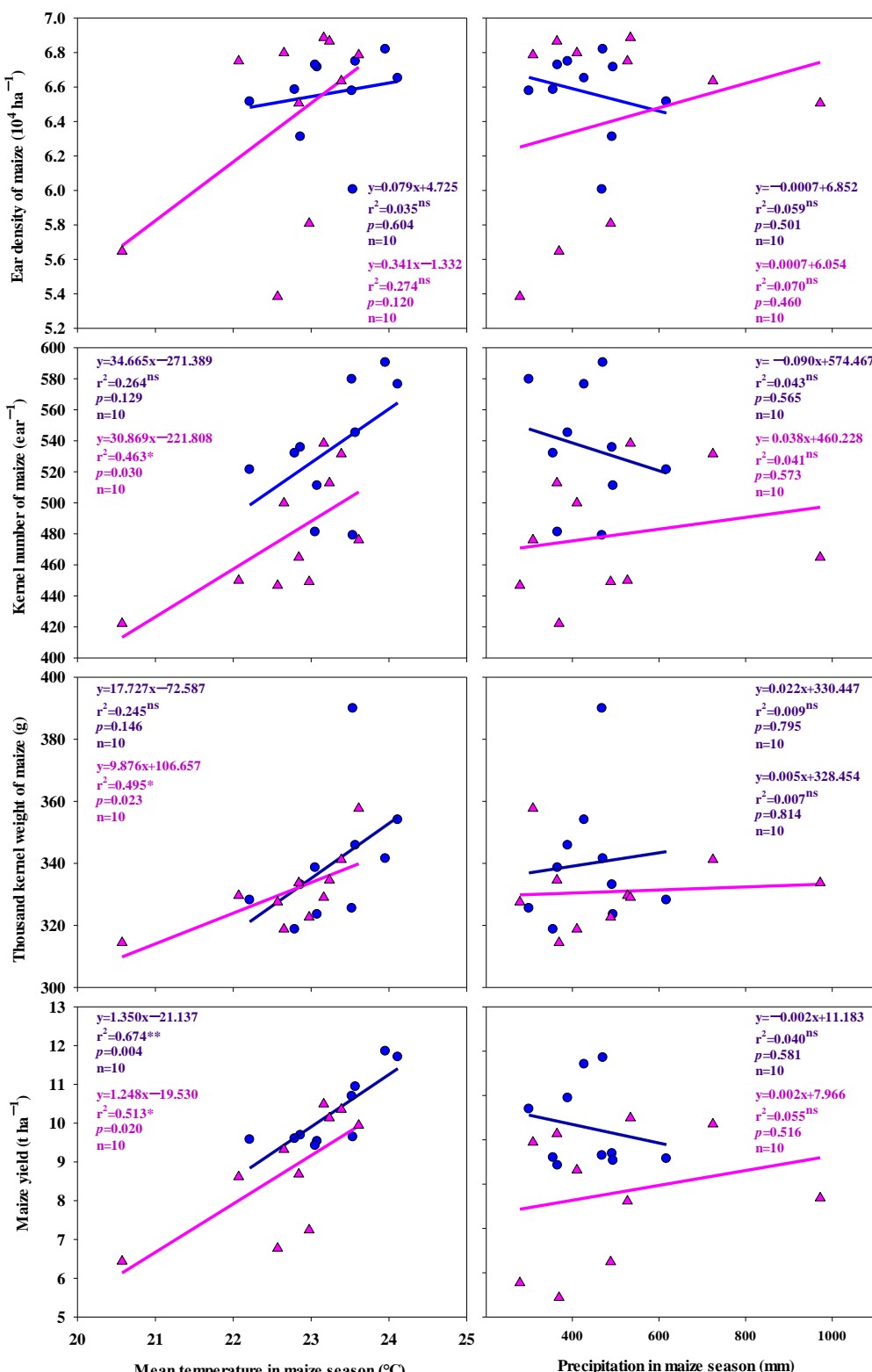

**Figure 10.** Correlation between maize yield, yield components and mean temperature and precipitation in the maize season. The blue circles, lines, and formulas represent the data of the Tai'an experimental site, and the pink circles and lines represent the data of the Longkou experimental site. r: correlation coefficient; *p*: significance level; n: sample size; **: significant at *p* < 0.01; *: significant at *p* < 0.05. ns: not significant.

### 3.6. The Relationship between Climate and Crop Yield Trends

Climate and crop yield trends were the values of the following year minus the previous year. At the warm-dry site, the maize yield trend was obviously correlated with the mean temperature trend in the maize season (Figure 11). At the cool-wet site, the maize yield trend was obviously correlated with the mean temperature trend at the 0.01 level in the maize season. The wheat yield trend had no obvious correlation with the mean temperature and precipitation trends, and the maize yield trend had no apparent divergence from the mean precipitation trend.

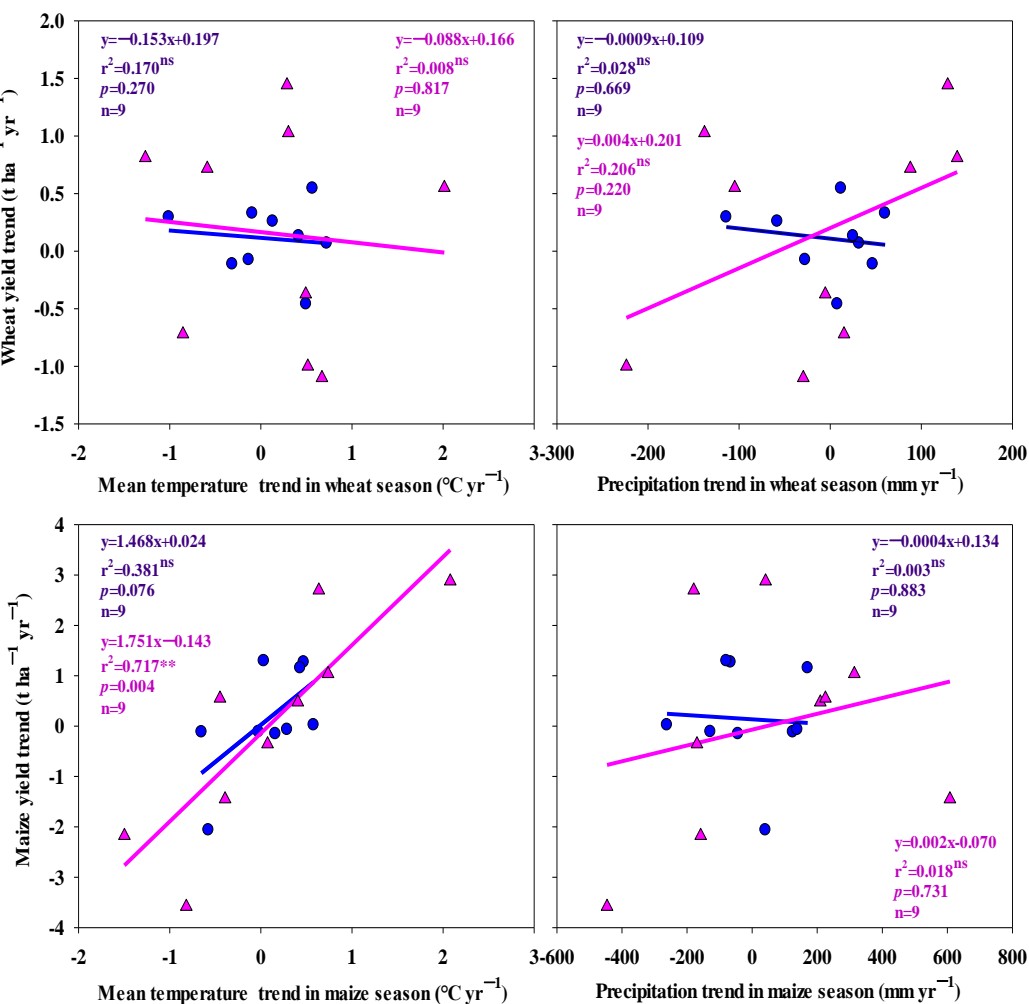

**Figure 11.** Correlation between wheat and maize yield and mean temperature trend and precipitation trend. The trend represents the difference of the following year minus the previous year. The blue circles and lines represent the data of the Tai'an experimental site, and the pink circles, lines, and formulas represent the data of the Longkou experimental site. r: correlation coefficient; *p*: significance level; n: sample size; **: significant at $p < 0.01$; ns: not significant.

### 3.7. Structural Equation Modeling Analysis

Structural equation modeling further revealed the direct and indirect effects of mean temperature, annual precipitation, and yield components on crop yield (Figure 12). The combination of mean temperature, annual precipitation, and yield components explained 75% and 100% of the variance in the wheat yield and maize yield, respectively. The mean temperature and annual precipitation had weak direct effects but stronger indirect effects on crop yield through their associations with yield components. The mean temperature had a significant positive correlation with the ear density and thousand-kernel weight of wheat and with the ear density, kernel number, and thousand-kernel weight of maize. The annual precipitation had no significant correlation with the yield components, which was

inconsistent with our hypothesis. The ear density of wheat had a strong negative effect on thousand-kernel weight. The thousand-kernel weight played a major role in regulating wheat yield, and the kernel number played a major role in regulating maize yield.

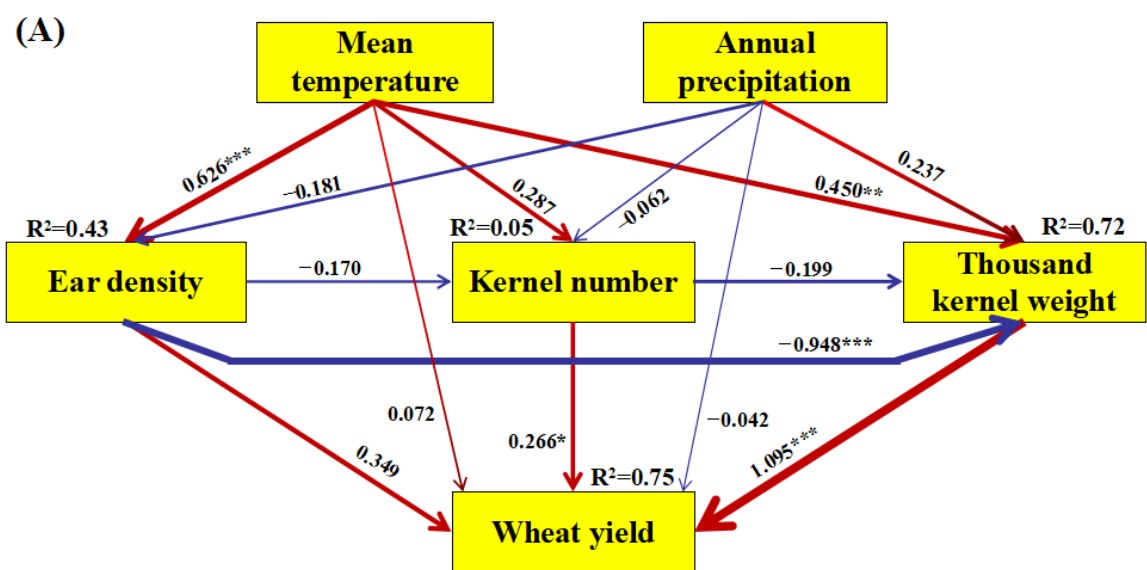

$\chi^2=0.720$, $P=0.396$, GFI=0.988, CFI=1.000, AIC=40.720, RMSEA<0.001

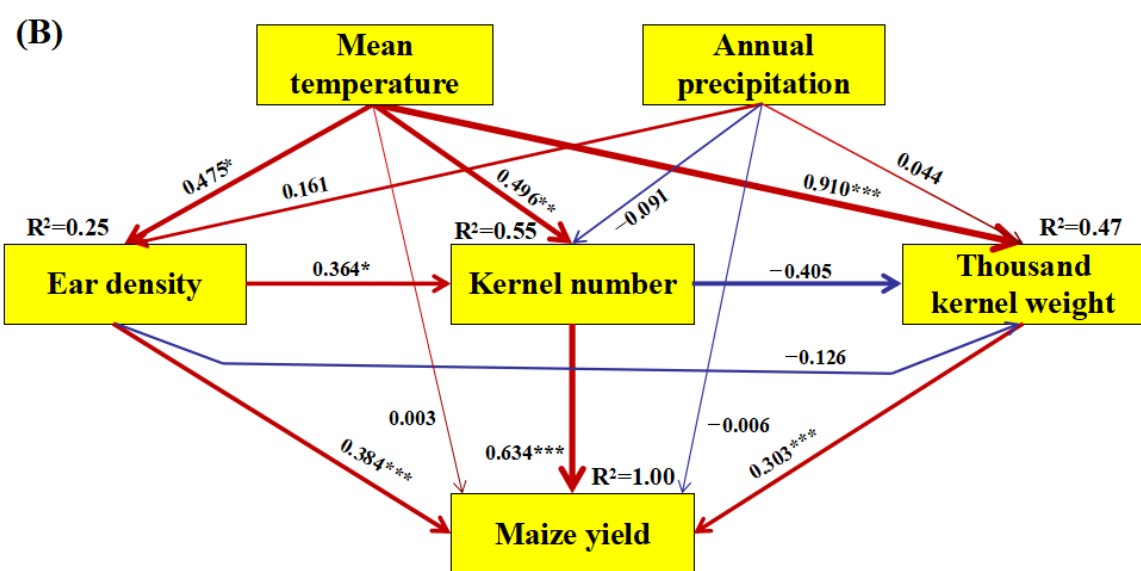

$\chi^2=0.022$, $P=0.883$, GFI=1.000, CFI=1.000, AIC=40.022, RMSEA<0.001

**Figure 12.** Direct and indirect effects of mean temperature, annual precipitation, and yield components on crop yield. Structural equation modeling was conducted for wheat yield (**A**) and maize yield (**B**). Single-headed arrows indicate the hypothesized direction of causation. Red and blue arrows indicate positive and negative relationships, respectively. The arrow width was proportional to the strength of the relationship. The numbers adjacent to the arrows are the standardized path coefficients. Significant differences are indicated: *, $p<0.05$; **, $p<0.01$; ***, $p<0.001$.

## 4. Discussion

### 4.1. Effects of Tillage Practices and Straw Management on Crop Yield and Components

The ear density and kernel number of wheat distinguish the sink stage from the sowing and anthesis stages. Minimum-tillage or no-tillage conditions were shown to increase the ear density and kernel number [39]. However, there is research that shows the opposite

result [40]. In our study, the yields of wheat and maize under rotary tillage at the warm-dry site were lower than those under conventional tillage, mainly because of the higher ear density under conventional tillage, which played a major role. CTS had the best effect on crop yield because the thousand-kernel weight in CTS was higher than that of CT, while RT had the worst effect. At the cool-wet site, the yields of wheat and maize under rotary tillage were higher than those under conventional tillage. Rotary tillage mainly increased the ear density of wheat, and the kernel number played a major role in the maize yield. RTS had the best effect on crop yield and yield components because the kernel number in RTS was higher than that in RT, while CT had the worst effect. However, no significant differences in yield components were observed between RTS and CTS or between RT and CT. In warm-dry and cool-wet conditions, tillage methods showed opposite effects on crop yield and yield components. The yield stability of wheat was higher than that of maize because the stability of the kernel number was better in long-term planting. Moreover, the difference in yield and yield components between treatments in the cool-wet treatment was less than that in the warm-dry treatment. However, there was no significant difference in yield stability between wheat and maize in different treatments. This may be because tillage and straw returning can increase soil organic carbon, which is conducive to the growth of plant roots, increasing the ability to absorb water and nutrients, thus stabilizing and improving crop growth [41]. The improvement of agricultural productivity depends on the local environment and soil conditions [42]. In Europe, the implementation of conservation tillage may indeed reduce yield, which is related to tillage technology, rotation, crop type, and soil texture [43].

Different tillage methods are important for increasing crop yield. Minimum tillage or no-tillage combined with straw return can increase crop yield and reduce environmental pollution and soil degradation [44,45]. Compared with conventional tillage, the wheat and maize yields increased by 12.3–16.9% under minimum tillage combined with straw return [46]. Reduced or no-tillage alone cannot improve crop yield and yield stability [47]. After long-term wheat and maize cropping, rotary tillage alone cannot produce good effects. This is because long-term rotary tillage formed a plow bottom at a depth of 10 cm, the tractor pulls the rotary cultivator to cultivate the land, and the subsequent process of sowing, fertilization, and harvesting makes the soil below 10 cm mechanically compacted, and hindering root elongation and reducing crop yield. Crops under conventional tillage can grow more fine roots [48], thus enhancing yield growth. Straw return increased crop yield and yield components compared with no straw return. A global meta-analysis showed that straw returning can increase soil organic matter by 12.8% [49]. A previous study has also shown that crop yields increase as soil organic matter increases [50]. However, in the long-term experiment, straw return did not continuously show a significantly higher effect on crop yield than no straw return. Therefore, there was no need to return straw to the field every year after several years, and the excess straw can be used for feed, energy, and other purposes [51].

In addition, the production costs and energy consumption of different tillage practices and straw use are also very important to evaluate the yield and components of wheat and maize. In this study, production inputs include mechanical oil consumption such as tilling (28.9 USD ha$^{-1}$ for CT and 17.5 USD ha$^{-1}$ for RT), sowing (35.4 USD ha$^{-1}$), stubble control (36.6 USD ha$^{-1}$), pesticide spraying (12.8 USD ha$^{-1}$), and harvesting (59.8 USD ha$^{-1}$), and other costs such as seed (86.6 USD ha$^{-1}$), irrigation (0.3 USD ha$^{-1}$), fertilization (571.4 USD ha$^{-1}$), pesticide (103 USD ha$^{-1}$), and labor (107.1 USD Person$^{-1}$ ha$^{-1}$). The results showed that the production cost of rotary tillage was 1.1% lower than that of conventional tillage.

### 4.2. Effects of Temperature and Precipitation on Crop Yield and Components

The ear density and kernel number of wheat in warm-dry conditions were 16.1% and 0.9% higher than those in cool-wet conditions, and the thousand-kernel weight and yield were 17.9% and 16.4% lower, respectively. The thousand-kernel weight of maize

in warm-dry conditions was 1.9% lower than those in cool-wet conditions, and the ear density, kernel number, and yield of maize were 3.1%, 9.5%, and 9.8% higher, respectively. This result indicated that there was a certain regional effect of tillage that was due to the different climatic conditions. In the wheat season, the temperature at the warm-dry site was higher than that at the cool-wet site, and precipitation at the cool-wet site was greater than that at the warm-dry site. In the maize season, cool-wet maize had wider temperature and precipitation range values than warm-dry maize. The temperature and precipitation trend range values at the cool-wet site were greater than those at the warm-dry site.

Wheat yield is greatly influenced by climatic factors, such as temperature and precipitation. In rainy years, the harvest may be good, but there will be serious yield loss because either the crops cannot be harvested or the grains cannot be dried [52]. The wheat yield decreases with higher temperature but increases with more precipitation [53]. In this study, temperature and precipitation had weak positive and negative effects on wheat yield, respectively (Figure 10). From heading to anthesis, if the precipitation is less, the ear density of wheat will be seriously reduced [54]. This study showed that compared with 229.3 mm precipitation, 76 mm precipitation reduced the ear density, kernel number, and thousand-kernel weight and then reduced the wheat yield by 65%. Compared with other years, the lowest precipitation occurred at the anthesis stage of wheat in 2019, which led to a sharp decrease in ear density at the warm-dry site. For summer maize, the precipitation decreased by 14.1%, the ear density, kernel number, and thousand-kernel weight decreased, and the yield decreased by 45.2%. The kernel number was the main reason for the decrease in summer maize yield [46], which is consistent with the results of this study (Figure 10). At the warm-dry site, the highest kernel number and yield of maize were achieved in 2016 due to relatively higher precipitation during the anthesis stage. In 2018 and 2019, however, although the thousand-kernel weight was higher, the ear density and kernel number were the lowest because of heavy precipitation, which resulted in the lowest yield. At the cool-wet site, the low maize yield and yield components in 2011 may be attributed to the low temperature during the filling stage. Thus, temperature was a major limiting factor in cool-wet crop yields.

An assessment of the impact of seasonal climate change on interannual wheat and maize yields in 92 regions of France revealed regional differences in the most important variables affecting crop yields [18]. A temperature increase of 0.5 °C in winter may reduce India's wheat production by 0.45 t ha$^{-1}$ [55]. A 1 °C rise in temperature could reduce global wheat production by 6% [56]. The global maize yields decreased by 7.4% for each degree increase in the global mean temperature [57]. In this study, an approximately 38.1% maize trend was indicated by the mean temperature. When the temperature increased by 1 °C, the maize yield increased by 1.5 t ha$^{-1}$ at the warm-dry site. Approximately 71.7% of maize trends can be indicated by the mean temperature. When the temperature increased by 1 °C, the maize yield increased by 1.8 t ha$^{-1}$ at the cool-wet site. Therefore, drought-related temperature anomalies led to a significant decline in maize yield. All crops have a temperature threshold, and the yield will decrease with increasing temperature when it exceeds the threshold [58]. A study reported that maize and soybean yields declined by 17% for a 1 °C increase in temperature [59].

## 5. Conclusions

After ten years (2010–2019) of tillage and straw return to the field, CTS and RTS were more effective in increasing wheat and maize yields than RT and CT at the warm-dry and cool-wet sites. The yield of rotary tillage at the warm-dry site was lower than that in conventional tillage because of the lower ear density of wheat and maize. However, the yield of rotary tillage at the cool-wet site was higher than that in conventional tillage because of the higher ear density of wheat and higher kernel number of maize. The effects of straw return and site on crop yield and yield components were more obvious than tillage practices. Because the stability of the kernel number was better, the yield stability of wheat was higher than that of maize at the two sites. In the long-term experiment, straw return

did not continuously show a higher effect on crop yield than no straw return. Therefore, it can be considered that there was no need to return straw to the field year after year.

In terms of long-term results, wheat and maize yields and yield components in both regions were not always significantly correlated with temperature and precipitation. The explanatory degree of the combination of mean temperature, annual precipitation, and yield components on maize yield was 25% higher than that of wheat yield. The mean temperature and annual precipitation had the highest explanatory degree for the thousand-kernel weight of wheat and kernel number of maize. Moreover, the thousand-kernel weight and kernel number also played the most important roles in wheat yield and maize yield, respectively. Approximately 38.1% and 71.7% of maize trends can be indicated by the mean temperature; when the temperature increased by 1 °C, the maize yield increased by 1.5 and 1.8 t ha$^{-1}$ at the warm-dry and cool-wet sites, respectively.

**Author Contributions:** Conceptualization, T.N.; methodology, Z.L.; software, Z.L., N.W. and L.W.; validation, N.W., J.L. and L.W.; formal analysis, Z.L., G.L. and T.N.; investigation, Z.L., N.W. and J.L.; resources, G.L. and T.N.; data curation, Z.L.; writing—original draft preparation, Z.L.; writing—review and editing, Z.L.; visualization, Z.L. and L.W.; supervision, T.N.; project administration, Z.L. and T.N.; funding acquisition, Z.L. and T.N. All authors have read and agreed to the published version of the manuscript.

**Funding:** This research was funded by the National Natural Science Foundation of China (Grant No. 32101853), the Natural Science Foundation of Shandong Province (Grant No. ZR2021QC189), the Major Science and Technology Innovation Projects of Shandong Province (Grant No. 2019YQ014 and 2021CXGC010804-05), and the Funds of Shandong "Double Tops" Program.

**Data Availability Statement:** The data presented in this study are available on request from the corresponding author.

**Conflicts of Interest:** The authors declare no conflict of interest.

### Abbreviations

CT: conventional tillage without straw returning; CTS, conventional tillage with straw returning; RT, rotary tillage without straw returning; RTS, rotary tillage with straw returning; SYI, the yield stability index; L, experimental site; T, tillage practice; S, straw management.

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
