# Peer review of "Climate-Smart Tillage Practices with Straw Return to Sustain Crop Productivity"

_agronomy, doi:10.3390/agronomy12102452_

Round 1
Reviewer 1 Report
Please read your abstract carefully, and correct the way of presentation of abbreviations.
The hypothesis and the objectives are not well defined. Please make a good story and start it from the abstract itself by linking every sentence.
While the study is on tillage and residue retention but the authors have explained more about the water stress and other abiotic stresses.
The yield stability was reported in the case of maize as well as wheat, but in discussion part not adequately explained the reasons for yield stability for these crops due to tillage and straw mulch retention.
The findings like "The annual precipitation had no significant correlation with the yield components." This is self-contradictory from the introductory remarks by the authors.
The concluding statement in the abstract and in the conclusions are contradicting, like which is more impactful, tillage or straw mulch retention.
The references also need revision, please follow the MDPI guidelines and revise the presentation of the references.

Author Response
Dear reviewer,
Thank you for your valuable comments and suggestions on our manuscript. We write this Response to Comments in order to explain the queries of comments. Hope, these supplements as well as current version of the manuscript satisfy the queries and consider it by your journal.
Thank you very much again.
Sincerely,
Tangyuan Ning
Reviewer 1 (with pdf)
- Please read your abstract carefully, and correct the way of presentation of abbreviations.
--We have corrected the way of presentation of abbreviations.
“Conventional tillage (CT) and rotary tillage (RT) in combination with no straw return and whole straw return (S), were conducted under a wheat–maize cropping system in cool-wet and warm-dry regions from 2010–2019. Conventional tillage with straw return (CTS) in the warm-dry region and rotary tillage with straw return (RTS) in the cool-wet region can increase the yield and yield components of wheat and maize, respectively, compared with CT.” (L15-22)
- weather aberrations, or climatic vulnerabilities
--We have replaced “climatic conditions” with “climatic vulnerabilities”. (L42)
- not for all crops, also have vice versa effect.
“Globally, there is a negative correlation between climate warming and some crop productivity, such as wheat, maize and barley [11].” (L46-47)
- may be for more crops, and its more for wheat crop
--We have restated this sentence.
Precipitation was identified as the most important determinant of many crop yields [13]. (L48-49)
- its farming system or cropping system, please be clear while writing
--We have revised it.
“As a result, the effectiveness of cropping systems is highly geographically specific, and the impact of yield constraints can vary greatly depending on environmental conditions and their interactions with management practices” (L57-60)
- Its cropping pattern or cropping system, please clarify.
--We have revised it.
“Winter wheat-summer maize rotation is the most typical cropping system on the North China Plain” (L64-65)
- The hypothesis and the objectives are not well defined. Please make a good story and start it from the abstract itself by linking every sentence.
--We have restated the hypothesis and the objectives in the Abstract and Introduction.
“The objectives of this study were to identify and compare the differences of crop yield and yield components in long-term tillage and straw returning under different climate regions. Conventional tillage (CT) and rotary tillage (RT) in combination with no straw return and whole straw return (S), were conducted under a wheat–maize cropping system in cool-wet and warm-dry regions from 2010–2019. We hypothesized that long-term suitable tillage under warm-dry or cool-wet regions can increase the yield and components of wheat and maize, and temperature and precipitation had significant effects on crop yield and yield components.” (L13-19)
“The objectives of this study were:(1) to identify the differences of wheat and maize yields and yield components in long-term tillage and straw returning; (2) to compare the yield and yield component stability of wheat and maize under tillage and straw returning practices; (3) to clarify the correlation between crop yield, yield components and mean temperature and precipitation. It was hypothesized that long-term suitable tillage under warm-dry or cool-wet regions can increase the yield and components of wheat and maize, and temperature and precipitation had significant effects on crop yield and yield components.” (L90-98)
- While the study is on tillage and residue retention but the authors have explained more about the water stress and other abiotic stresses.
--We have deleted it according to your suggestion.
- Data is not properly reported, the quantification of the effect due to treatments imposition has not been properly reported.
--We have restated this section.
“Climate and crop yield trends were the values of the following year minus the previous year. At the warm-dry site, the maize yield trend was obviously correlated with the mean temperature trend in the maize season (Figure 11). At the cool-wet site, the maize yield trend was obviously correlated with the mean temperature trend at the 0.01 level in the maize season. The wheat yield trend had no obvious correlation with the mean temperature and precipitation trends, and the maize yield trend had no apparent divergence from the mean precipitation trend.” (L304-310)
- The yield stability was reported in the case of maize as well as wheat, but in discussion part not adequately explained the reasons for yield stability for these crops due to tillage and straw mulch retention.
--We have explained the reasons for yield stability for these crops due to tillage and straw mulch retention.
“The yield stability of wheat was higher than that of maize because the stability of kernel number was better in long-term planting. Moreover, the difference in yield and yield components between treatments in the cool-wet treatment was less than that in the warm-dry treatment. However, there was no significant difference in yield stability between wheat and maize in different treatments. This may be because tillage and straw returning can increase soil organic carbon, which is conducive to the growth of plant roots, increasing the ability to absorb water and nutrients, thus stabilizing and improving crop growth [42]” (L341-348)
- The findings like "The annual precipitation had no significant correlation with the yield components." This is self-contradictory from the introductory remarks by the authors.
--We have modified and explained it.
“The annual precipitation had no significant correlation with the yield components, which was inconsistent with our hypothesis.” (L320-321)
- The concluding statement in the abstract and in the conclusions are contradicting, like which is more impactful, tillage or straw mulch retention.
--Both the abstract and the conclusion have pointed out that straw return was more effective on crop yield than tillage.
“Compared with tillage practices, the effects of experimental sites and straw return on crop yield and yield components were more remarkable.” (L23-24)
“The effects of straw return and site on crop yield and yield components were more obvious than tillage practices.” (L442-443)
- The references also need revision, please follow the MDPI guidelines and revise the presentation of the references.
--We have revised the references according to your suggestion.
Reviewer 2 Report
This is an interesting paper presenting the results of a long term continuous double cropping per year of wheat and maize experiment in two regions. Two tillage treatment and tow straw handling methods were compared. I think the paper can be improved by improving the description of the treatments the authors applied. A description of the machine that applied the rotary tillage would help understanding the conditions of the experiment. The same for the straw remaining in the field. How it was chopped and spread? The removal was done manually. The chopping? What was the size of the plots?
In the discussion, I think that crop yield and the related parameters are not the only factors that have to be considered. Production cost, energy use etc are equally important. In CT, as I understand, there are 3 passes with machinery (disking, ploughing and rotary cultivator) while in RT only a rotary cultivator. In terms of cost and energy use the difference is really important. I think the authors have to include this in their discussion.
Some comments to help improve the paper.
L 64-65 Over the past 50 years, 13% of maize production in the United States has been related to drought [24]. Is this correct? Is there a missing word?
L323 -324 This is because long-term rotary tillage formed a plow bottom at a depth of 10 cm, 323 which hindered root elongation and reduced crop yield.
Does a rotary cultivator cause soil compaction underneath tillage depth? There is a problem not describing the machinery used. From what I know rotary cultivator tends to enter into the soil and the working depth is defined by the cylinder following the rotating tines. So, please explain how soil compaction is caused. Plough compacts the soil as the downward force produced by the soil movement compacts the soil.
L 400-401 Therefore, it can be considered that there was no need to return straw to the field year after year.
What is the meaning of this statement? The variation of yield apart from straw retention comes from weather conditions. As there is not forecast for a year how we can use this? One question worth discussing is what is the effect of straw incorporation in the soil? Is this an increase of soil organic matter? Or something else? The high soil disturbance could increase soil organic matter decay but what is the final result. Is there a long term effect of possible increase of soil organic matter in the yield?
Author Response
Dear reviewer,
Thank you for your valuable comments and suggestions on our manuscript. We write this Response to Comments in order to explain the queries of comments. Hope, these supplements as well as current version of the manuscript satisfy the queries and consider it by your journal.
Thank you very much again.
Sincerely,
Tangyuan Ning
Reviewer 2
This is an interesting paper presenting the results of a long term continuous double cropping per year of wheat and maize experiment in two regions. Two tillage treatment and tow straw handling methods were compared. I think the paper can be improved by improving the description of the treatments the authors applied.
--Thank you very much for your careful and detail comments and suggestions on the paper.
A description of the machine that applied the rotary tillage would help understanding the conditions of the experiment. The same for the straw remaining in the field. How it was chopped and spread? The removal was done manually. The chopping? What was the size of the plots?
--We have added these descriptions in the Materials and Methods section.
Three replicates were set for each treatment, and the size of each plot was 15×4 m2 at Tai’an and 15×15 m2 at Longkou. The operational procedures for conventional tillage were: maize was harvested mechanically, and straw was crushed (3–5 cm long) and returned to the field at the same time for straw return treatment (or after harvesting the maize manually, cut the straw against the ground and move it out of the farmland for no straw return treatment)–fertilizer application–stubble plowing with notched harrow–plowing with moldboard plow (25 cm)–rotary tillage with rotavator (10 cm)–wheat sowing–irrigation–herbicide application. The operational procedures for rotary tillage were: maize was harvested mechanically, and straw was crushed (3–5 cm long) and returned to the field at the same time for straw return treatment (or after harvesting the maize manually, cut the straw against the ground and move it out of the farmland for no straw return treatment)–fertilizer application–stubble plowing with notched harrow–rotary tillage with rotavator (10 cm)–wheat sowing–irrigation–herbicide application. Rotary cultivator is an agricultural machine that breaks the soil by rotating the blade teeth. Tractors are often used as power sources for rotary cultivators to provide power for driving and operation. (L126-141)
In the discussion, I think that crop yield and the related parameters are not the only factors that have to be considered. Production cost, energy use etc are equally important. In CT, as I understand, there are 3 passes with machinery (disking, ploughing and rotary cultivator) while in RT only a rotary cultivator. In terms of cost and energy use the difference is really important. I think the authors have to include this in their discussion.
--We have added the production cost, energy use etc in the discussion.
“In addition, the production costs and energy consumption of different tillage practices and straw use were also very important to evaluate the yield and components of wheat and maize. In this study, production inputs include mechanical oil consumption such as tilling (28.9 $ ha-1 for CT and 17.5 $ ha-1 for RT), sowing (35.4 $ ha-1), stubble control (36.6 $ ha-1), pesticide spraying (12.8 $ ha-1) and harvesting (59.8 $ ha-1), and other costs such as seed (86.6 $ ha-1), irrigation (0.3 $ ha-1), fertilization (571.4 $ ha-1), pesticide (103 $ ha-1) and labor (107.1 $ Person-1 ha-1). The results showed that the production cost of rotary tillage was 1.1% lower than that of conventional tillage.”
(L377-384)
Some comments to help improve the paper.
L 64-65 Over the past 50 years, 13% of maize production in the United States has been related to drought [24]. Is this correct? Is there a missing word?
--We have deleted this part.
L323-324 This is because long-term rotary tillage formed a plow bottom at a depth of 10 cm, which hindered root elongation and reduced crop yield.
Does a rotary cultivator cause soil compaction underneath tillage depth? There is a problem not describing the machinery used. From what I know rotary cultivator tends to enter into the soil and the working depth is defined by the cylinder following the rotating tines. So, please explain how soil compaction is caused. Plough compacts the soil as the downward force produced by the soil movement compacts the soil.
--Because the depth of the rotary tiller in this study is 10 cm, it can only turn the topsoil. After long-term rotary tillage, the soil below 10 cm is not disturbed and is pressed mechanically, resulting in soil compaction.
“This is because long-term rotary tillage formed a plow bottom at a depth of 10 cm, the tractor pulls the rotary cultivator to cultivate the land, and the subsequent process of sowing, fertilization and harvesting makes the soil below 10 cm be mechanically compacted, and hindered root elongation and reduced crop yield.” (L359-362)
L 400-401 Therefore, it can be considered that there was no need to return straw to the field year after year.
What is the meaning of this statement? The variation of yield apart from straw retention comes from weather conditions. As there is not forecast for a year how we can use this? One question worth discussing is what is the effect of straw incorporation in the soil? Is this an increase of soil organic matter? Or something else? The high soil disturbance could increase soil organic matter decay but what is the final result. Is there a long term effect of possible increase of soil organic matter in the yield?
--By this we mean that the effect of straw return and no straw return on crop yield is not significantly different in many years. Therefore, measures can be taken to return straw to the field every other year or part of the straw to the field, so that the straw can be used more efficiently. We have also added some discussion about the effects of straw returning on soil organic matter.
“A global meta-analysis showed that straw returning can increase soil organic matter by 12.8% [50]. Previous study has also shown that crop yields increase as soil organic matter increases [51]. However, in the long-term experiment, straw return did not continuously show a significantly higher effect on crop yield than no straw return.” (L365-367)
Round 2
Reviewer 1 Report
The authors have tried to include all the corrections, but for clarity please provide the data alongwith figures. From the figures it not easily understable what was the real impact of the treatments. Also the treatments effect is also ambiguous, therefore please provide data in table form at least for seed yield.
Coefficient of correlation and determination and regression are loosely used , plesse specify the terminology. There are figures the effect remains non significant, these data have least value may be given least priority or the authors can justify suitable like in Fig 9 and 10
Author Response
Reviewer 1
- The authors have tried to include all the corrections, but for clarity please provide the data along with figures. From the figures it not easily understandable what was the real impact of the treatments. Also the treatments effect is also ambiguous, therefore please provide data in table form at least for seed yield.
--We have added a table on crop yield.
Table 1. Mean yield of wheat and maize under different tillage and straw returning from 2010 to 2019.
|
Warm-dry site |
|
Cool-wet site |
|
|
Wheat yield (t ha-1) |
Maize yield (t ha-1) |
Wheat yield (t ha-1) |
Maize yield (t ha-1) |
CT |
7.7 a |
10.0 b |
8.1 a |
8.4 b |
CTS |
7.9 a |
11.0 a |
8.6 a |
8.9 ab |
RT |
7.0 b |
9.7 b |
8.3 a |
8.7 ab |
RTS |
8.2 a |
10.3 ab |
8.7 a |
9.3 a |
CT: conventional tillage without straw returning; CTS: conventional tillage with straw returning; RT: rotary tillage without straw returning; RTS: rotary tillage with straw returning. Different letters in each column indicate significant differences between different treatments (p<0.05; Duncan’s test).
- Coefficient of correlation and determination and regression are loosely used, please specify the terminology. There are figures the effect remains no significant, these data have least value may be given least priority or the authors can justify suitable like in Fig 9 and 10.
--We have specified the terminology in Figure 9, 10, 11 and Table 3.